# Genome of Alaskapox Virus, a Novel Orthopoxvirus Isolated from Alaska

**DOI:** 10.3390/v11080708

**Published:** 2019-08-01

**Authors:** Crystal M. Gigante, Jinxin Gao, Shiyuyun Tang, Andrea M. McCollum, Kimberly Wilkins, Mary G. Reynolds, Whitni Davidson, Joseph McLaughlin, Victoria A. Olson, Yu Li

**Affiliations:** 1Poxvirus and Rabies Branch, Division of High-Consequence Pathogens and Pathology, National Center for Emerging and Zoonotic Infectious Diseases, Centers for Disease Control and Prevention, Atlanta, GA 30329, USA; 2Alaska Division of Public Health, Section of Epidemiology, Anchorage, AK 99503, USA

**Keywords:** orthopox, alaskapox, poxvirus, orthopoxvirus, Alaska, phylogenetics, recombination

## Abstract

Since the eradication of smallpox, there have been increases in poxvirus infections and the emergence of several novel poxviruses that can infect humans and domestic animals. In 2015, a novel poxvirus was isolated from a resident of Alaska. Diagnostic testing and limited sequence analysis suggested this isolate was a member of the *Orthopoxvirus* (*OPXV*) genus but was highly diverged from currently known species, including *Akhmeta virus*. Here, we present the complete 210,797 bp genome sequence of the Alaska poxvirus isolate, containing 206 predicted open reading frames. Phylogenetic analysis of the conserved central region of the genome suggested the Alaska isolate shares a common ancestor with Old World OPXVs and is diverged from New World OPXVs. We propose this isolate as a member of a new OPXV species, *Alaskapox virus* (*AKPV*). The AKPV genome contained host range and virulence genes typical of OPXVs but lacked homologs of C4L and B7R, and the hemagglutinin gene contained a unique 120 amino acid insertion. Seven predicted AKPV proteins were most similar to proteins in non-OPXV Murmansk or NY_014 poxviruses. Genomic analysis revealed evidence suggestive of recombination with Ectromelia virus in two putative regions that contain seven predicted coding sequences, including the A-type inclusion protein.

## 1. Introduction

*Poxviridae* is a family of large, double-stranded DNA viruses that infect a broad range of animal hosts, from insects to vertebrates. The *Orthopoxvirus* genus contains some of the best-characterized poxviruses, including Variola virus (VARV, the causative agent of smallpox) and Vaccinia virus (VACV, the principal source of the smallpox vaccine). Orthopoxviruses (OPXVs) differ greatly in their host range and virulence [1,2,3]. Some OPXVs are specialists and only infect a single host species, such as VARV, which only infects humans [3]. Other OPXVs such as Cowpox virus (CPXV), VACV and Monkeypox virus (MPXV) have broad host ranges that include many species of mammals [3]. These zoonotic poxviruses can cause infections in humans and livestock [3] and represent potential threats to public health and substantial economic losses. Despite a wealth of information on a few OPXV species, little is known about the natural reservoir, host range and geographic distribution of the majority of OPXVs [2].

Cases of human infection by OPXVs have been increasing in recent years, with increases in cases of cowpox in Europe [4,5], outbreaks of monkeypox in African countries and North America [6,7,8,9,10,11], VACV infections in South America [12,13,14,15] and recent importations of monkeypox to the United Kingdom, Israel, and Singapore [16,17,18]. This increase in OPXV infections is thought to be due, at least in part, to waning population immunity caused by discontinuation of routine smallpox vaccination after the eradication of smallpox [7]. Smallpox vaccination provides protection against other *OPXV* species [19,20], and the increasing worldwide population who have never been vaccinated could provide the opportunity for emergence or reemergence of OPXV infections in humans [21]. 

Recently, several novel poxviruses have been discovered after infecting humans and/or domestic animals [22,23,24,25,26,27,28,29,30,31], including three newly proposed *OPXV* species: two isolated in Europe and one isolated in North America. Orthopoxvirus Abatino (OPVA) was isolated in Italy during an outbreak in captive macaques in 2015 and from a fatal infection in a cat in 2017 [28,29,31]. Another novel proposed OPXV, Ahkmeta virus (AKMV), was identified in three humans in the country of Georgia in 2010 and 2013 [22,27]. In 2015, a novel poxvirus was isolated from a resident of Alaska [30]. Phylogenetic analysis of several highly conserved genes suggested that the Alaska isolate was a member of the *OPXV* genus but was highly diverged from known *OPXV* species [30]. 

The *OPXV* genus contains two distinct clades of viruses in which genomic sequence similarity correlates to their historic geographic distribution in either the Old World or New World. Old World OPXVs contain seven currently recognized species, including *VARV*, *VACV*, *MPXV*, *CPXV*, *Camelpox virus*, *Ectromelia virus* (*ECTV*), and *Taterapox virus* [32], although the diversity of CPXV isolates suggests that *CPXV* may encompass several species [33,34]. There are currently three known species of New World or North American OPXVs: *Raccoonpox virus* (*RCNV*), *Volepox virus* (*VPXV*) and *Skunkpox virus* (*SKPV*) [32,35]. Newly described OPVA [29] and AKMV [27] cluster with Old World OPXVs in phylogenetic analyses, in agreement with their isolation in Europe, although AKMV is divergent [27]. The novel poxvirus isolated from Alaska in 2015 clustered with OPXVs, but formed a distinct clade, apart from Old World and New World OPXVs, based on preliminary analysis of nine conserved genes [30]. The nine genes of the Alaska isolate exhibited notable divergence from both New World and Old World OPXVs, with higher sequence similarity to Old World OPXVs [30], which was unexpected given its isolation in the New World. 

The divergence of the Alaska isolate and the seeming disagreement between its isolation in the New World and sequence similarity to Old World OPXVs highlighted a need for further investigation. Here, we present the whole genome sequence of the poxvirus isolated from an Alaska resident [30] and propose this isolate as a member of a new *OPXV* species, *Alaskapox virus* (*AKPV*).

## 2. Materials and Methods

### 2.1. Genome Sequences Used for Analysis

Accession numbers for reference OPXV genomes used in this study are listed in Appendix A. The genome sequence of the Alaskapox virus isolate described here was deposited with the GenBank accession number MN240300.

### 2.2. NGS Sequencing, De Novo Assembly and Gap Filling

Sample details and collection information was described previously [30]. The AKPV isolate was sequenced and assembled as previously described [27]. NGS sequencing was performed on an Illumina Hiseq 2000 at Otogeneitcs (Norcross, GA, USA). Reads were assembled using CLC Genomics Workbench 8.0 (Qiagen, Aarhus, Denmark). Output contigs were screened for poxvirus matches using BLAST. A draft genome was constructed that contained three gaps in the central region and incomplete ITR regions. Three central gaps were located around 174K, 163K, and 150K. Primer sets used for filling these gaps were, N1F (5′-CAT CTA CCA GAG AAA AAT GCG-3′) and G1R (5′- GGT GAT GCC GAA TAT TTC TAC -3′); G2F (5′-GAA AAG GAG ATA GTG ATT GTC-3′) and G2R (5′-CTA TCT TTG ACG TCG ATG TGG-3′); G3F (5′-CGA TCA GCG TCC TTT TTG-3′) and G3R (5′-TGT TAA TA AGTA ATA ACT GCG C-3′). Terminal tandem repeat sequences were determined using ER1F ( 5’-AGT GTC TAG AAA AAA ATG TGT GAC CGC-3’) and ER1R (5’-GGA TAC TGC TCA CGT TTT TT-3’). Sequencing reaction was performed using the BigDye^®^ Terminator v3.1 Cycle Sequencing Kit (ABI, Cat 4337035) on an ABI 3130XL Genetic Analyzer. Called sequences were analyzed using Seqman in the Lasergene Package (DNASTAR, Madison, WI, USA). Consensus sequences were used to close gaps in the draft genome. Raw reads were mapped back to the gap-filled genome to inspect assembly accuracy.

### 2.3. Gene Prediction and Annotation

Gene prediction was conducted using both an *ab initio* approach [36] and similarity search using GeneMarkS [37] and Geneious version 9.1.6 (Biomatters, Inc., Newark, NJ, USA). First, ORFs longer than 120 nt with ATG as the start codon were found in Geneious using the ORFfinder tool. Regions with >60% similarity to genes in any of seven Cowpox virus genomes were identified in the Alaska genome using the similarity-transfer functionality in Geneious. ORFs longer than 500 nt and ORFs with 20% in-frame overlap with the transferred regions were combined using a customized script. This set of ORFs was further combined with ORFs predicted by GeneMarkS as potential genes. Functional annotation of the genes was conducted in Geneious with the Blast2Go plugin using a customized poxviridae database. Annotations were then manually inspected, corrected and confirmed.

### 2.4. Alignment of Poxvirus Genomes

All alignments presented were generated using mafft v7.308 [38,39] in Geneious version 9.1.4 (Biomatters, Inc., Newark, NJ, USA) using the FFT-NS-ix1000 algorithm, 200 PAM/k = 2 scoring matrix with default gap penalties. Percent identity graphs were generated by calculating the average percent identity across 100 columns of the alignment and graphed using ggplot2 [40]. 

### 2.5. Phylogenetic Analysis of the Conserved Central region

The region from VACV-COP-E9L (DNA polymerase) to A24R (DNA-dependent RNA polymerase subunit rpo132) was extracted from AKPV, reference OPXV genomes, and NY_014, Murmansk, and Yoka poxvirus genomes (Appendix A) and aligned. Phylogenetic trees were generated in BEAST v.2.5.1 [41] in two runs with the following parameters: GTR+G+I nucleotide substitution model (4 gamma categories, 35% invariant, based on model test performed in MEGA 7.0.26), relaxed lognormal clock (exponential distribution of ucldStdev prior with mean = 0.333), and Yule model prior until all parameters exhibited ESS > 200 after 10% burn-in. Default parameters were used unless specified. Run log and tree files were combined using LogCombiner after 10% burn in. Maximum clade credibility tree was estimated in TreeAnnotator based on sampling frequency of 1000 and 10% burn-in; tree was visualized in FigTree v1.4.3 (https://github.com/rambaut/figtree/) and edited in Inkscape (https://inkscape.org/). Percent nucleotide identities were reported from Geneious version 9.1.4 (Biomatters, Inc., Newark, NJ, USA) based on alignment where all columns containing gaps were stripped.

### 2.6. Gene Content Comparison

A reciprocal blast approach was used to compare the AKPV genome with CPXV-BR and AKMV-2013. All predicted CDS were extracted from the AKPV, CPXV-BR, and AKMV-2013 genomes. CPXV-BR or AKMV-2013 CDS were queried against AKPV CDS, and AKPV CDS were queried against the CPXV-BR or AKMV-2013 CDS using blastn (ncbi-blast+ v2.4.0). Hits were determined using a cutoff Evalue of 0.01. To identify the closest annotated proteins for each predicted AKPV protein, all 206 predicted ORFs were extracted from the AKPV genome and translated into predicted amino acid sequences. The 206 predicted proteins were queried against the ‘nr’ database using blastp (ncbi-blast+ v2.4.0). Data reflects BLAST search performed on 12 February 2019. Top hit for each query sequence was determined based on bit score using default blastp parameters.

### 2.7. dN/dS Analysis

To investigate if poxvirus homologs of T4 and B5R were under positive selection in AKPV, dN/dS was examined using PAML [42]. T4 sequences from CPXV-A, CPXV-B, CPXV-C, CPXV-D, CPXV-E1, CPXV-E2, CPXV-E3, CPXV-E5, CPXV-E6, CMLV, MPXV-DRC, TATV, AKMV-2013, SKPV, and VPXV were used. The same genomes used for the T4 analysis plus ECTV-Moscow, MPXV-USA, HSPV, OPVA, VARV-IND, VARV-BRA, AKMV-2010, and RCNV-Herman were used for the B5R analysis. DNA polymerase, B5R or T4 CDS from AKPV and reference genomes were aligned using mafft v7.308 [38,39] in geneious version 9.1.4 using the FFT-NS-ix1000 algorithm, 200 PAM/k = 2 scoring matrix with default gap penalties. DNA polymerase CDS alignments were used to estimate phylogenetic trees by Maximum Likelihood in Mega 7.0.26 [43] using the GTR+G+I model and 1000 bootstrap replicates; substitution model was chosen based on a Model Test performed in Mega 7.0.26. 

B5R and T4 alignments were analyzed for evidence of selection in AKPV using branch-site models in the CODEML package in PAML 4.5 [42]. Log likelihood tests were performed to determine if a model that included codons under positive selection in AKPV (foreground) fit the alignment for a given gene better than a model that did not allow positive selection (comparison of model A to A1). Model A was specified by NSsites = 2, model = 2, fixomega = 0. Model A1 was specified by NSsites = 2, model = 2, fixomega = 1. Analyses were performed using a user tree (DNA polymerase), CodonFreq = 2, and all ambiguous sites were removed. Sites where dN/dS > 1 were identified based on Bayes Empirical Bayes (BEB) analysis [44] using Pr(ω > 1) > 0.95 as a cutoff.

### 2.8. Investigation of Potential Recombination Events

A region from AKPV143 (DNA helicase) to AKPV175 (hypothetical protein) (corresponding to position 138,271 to 166,556 of the AKPV genome) was aligned to corresponding regions from AKMV-2010, VARV-BRA, HSPV, CPXV-A, CPXV-C, CPXV-E3, TATV, CMLV, ECTV-Moscow, ECTV-Hamptead, ECTV-Naval, OPVA, VPXV, SKPV, and RCNV-85A reference genomes using MAFFT v7.308 [38,39] (algorithm: FFT-NS-I x1000, scoring matrix: 200PAM, k=2, gap open: 3, offset: 0.123) implemented in Geneious 9.1.4. Gaps were not removed from the alignment prior to recombination analysis. An exploratory search for potential recombination events was performed using bootscan and distance plot implemented in the RDP4 software using a window size of 1000, step size of 50, under Felsenstein 1984 [45] model using bootstrap value as P value. Data for bootscan and similarity plots were generated using RDP4; plots were generated from raw data using tidyr [46] and ggplot2 [40] in RStudio v.1.0.44(R version 3.3.3) [47,48]. Percent identity of aligned regions was calculated in Geneious 9.1.4. Phylogenetic analysis by Maximum Likelihood was performed in MEGA7 [43] based on the GTR+G+I model.

## 3. Results

### 3.1. Genome Characteristics

The complete genome of the AKPV isolate was 210,797 bp, with inverted terminal repeats (ITRs) of 2.4 kb. The genomic terminal loop was not sequenced, so the leftmost nucleotide was arbitrarily assigned to be the first nucleotide. The A+T content was 67.2%, lower than the A+T content of New World OPXVs such as VPXV (68.7%), SKPV (68.5%) and RCNV (67.7–67.9%), but higher than that of most Old World OPXVs (66.3–67.3%) [1,49]. In the right terminal region of the genome, a long stretch of homopolymer G could not be resolved by Sanger sequencing and is represented by 22 Ns (position 200,082 to 200,103).

The AKPV ITR was 2.4 kb, which is shorter than most OPXVs except VARV, where ITRs range from 0.1 to 1.2 kb [1]. Outside of VARV, ITRs in Old World OPXVs vary from 3.4 (VACV) to 16.4 kb (VACV) [1]. New World OPXVs ITRs also vary greatly in size from 2.5–3.9 kb for SKPV, VPXV, and RCNV-85A [35] to 19 kb in RCNV-Herman [50]. The AKPV ITR region contained tandem repeats and non-repetitive elements that were characteristic of OPXVs. The AKPV genome contained the concatemer resolution sequence 5ʹ-ATTTA-N79-A67-3ʹ, identical to that in VACV Copenhagen, at positions 61–80bp in the AKPV genome. A tandem repeat region from position 151–1525 contained 16 copies of an 86bp repetitive motif (98.3% identical). The 86bp repeat motif was 94% identical to the 86bp repeat sequence in AKMV tandem repeat region 1 (TR1), which is repeated 6.3 times in the AKMV ITR [27]. This tandem repeat region in AKPV is followed by a non-repetitive region (positions 1550–1703 bp) that is similar to NR2. The NR2-like sequence was 94% identical to positions 1704 to 1850 in CPXV-BR.

### 3.2. Phylogenetic Analysis of the Central Core Region

The central region of OPXV genomes is highly conserved and contains genes involved in essential processes such as transcription, DNA repair and replication [49,51,52]. This central region is flanked by variable terminal regions that contain host range, virulence, and immunomodulatory genes [49,51,52]. Gene content and synteny are highly conserved in the central region but can vary between species and strains in the terminal regions, presumably due to selection and recombination [49,53,54,55]. Phylogenetic analysis was performed using the central region (corresponding to VACV-COP-E9L (DNA polymerase) to A24R (DNA-dependent RNA polymerase subunit rpo132)) of AKPV, representative OPXVs and NY_014, Murmansk, and Yoka poxviruses (Appendix A). AKPV formed a distinct branch located between Old World and New World OPXVs, but closer to Old World OPXVs (Figure 1). The central region of the AKPV genome was more similar to Old World OPXVs than to New World OPXVs, with the highest nucleotide identity with AKMV (93.5%) (Table 1). The AKPV central region was, on average, 92.9% identical to the Old World OPXVs examined (ranging from 92.4–93.5%), compared to 87.1% average identity to New World OPXVs (range: 86.9–87.4%)(Table 1), reflecting its position as a sister branch in the phylogeny (Figure 1). However, the average identity within Old World OPXVs examined was 97.52% (95.1–99.7%), much higher than the identity of AKPV to Old World OPXVs.

### 3.3. Comparison of Gene Content

Gene annotation using both *ab inito* gene prediction and similarity search revealed 206 predicted genes (Appendix A). The AKPV genome was compared to the CPXV Brighton Red (CPXV-BR) genome, which is 224,499 bp and contains 229 predicted coding sequences (Appendix A). Overall, gene content and organization was similar between the AKPV and CPXV-BR genomes, including at the genomic termini (Appendix A). Twenty-three CPXV-BR genes were not found in the AKPV genome, and six predicted AKPV genes did not correspond to any annotated CPXV-BR genes, not including truncated or fragmented genes. Of the six unique AKPV genes, two (AKPV204 and 205) did not have homologs in any currently known OPXV. Five terminal genes were duplicated in the CPXV-BR genome but were found in single copies in the AKPV genome. For the 23 genes not found in the AKPV genome, fifteen are not annotated in the CPXV_E1 genome (CPXV004, 007, 047, 058, 096, 116, 130, 160, 170, 192, 214, 216, 224, 228, and 229). The 206 predicted AKPV coding sequences were then compared to annotated genes from CPXV-BR and AKMV-2013 (220 predicted coding sequences). 114 AKPV genes exhibited higher nucleotide identity to AKMV genes based on BLAST search; 82 were more similar to CPXV-BR genes (Appendix A). The average nucleotide identity of AKPV genes found in AKMV and CPXV-BR was 90.0% and 89.8%, respectively (Appendix A).

Each of the 206 AKPV predicted proteins were then used to query the non-redundant database to identify novel genes and compare AKPV proteins to those of other well-studied OPXV genomes. All 206 predicted proteins in the AKPV isolate genome exhibited >50% amino acid identity with an annotated poxvirus protein, with an average amino acid identity of 89.6% with the closest BLAST hit. 191 predicted proteins (92.7%) were most closely related to an Old World OPXV sequence, including AKMV and OPVA (Appendix A). 

Eight predicted AKPV proteins returned top hits to New World OPXV proteins, including AKPV011 (TNF alpha receptor-like protein), AKPV013 (ankyrin repeat-containing protein), AKPV100 (Poly(A) polymerase small subunit), AKPV102 (late 16 kDa membrane protein), AKPV112 (virion core protein), AKPV123 (trimeric virion coat protein), AKPV126 (S-S bond formation pathway protein) and AKPV191 (kelch-like protein) (Appendix A). AKPV011 was most similar to SKPV TNF receptor CrmB (gene 203 in the SKPV genome), while AKPV013 was most similar to RCNV (gene 011 in the RCNV Herman strain genome) (Table 2). However, SKPV and RCNV Herman strain genomes lack an intervening coding sequence, while Old World OPXV genomes contain homologs of AKPV011, 012, and 013 in order (Appendix A). AKPV011 and AKPV013 did not share high similarity with predicted proteins from VPXV or RNCV strain 85A (<50% and 60% amino acid identity, respectively, for the closest sequences in VPXV genome and 50% coverage and <50% identity with the closest sequences in RCNV stain 85A). The AKPV genome did not contain homologs of SKPV174/VPXV174/RCNV159 or RCNV160/SKPV175, genes that are thought to be unique to New World OPXVs.

The majority of the 206 predicted proteins (97%) were most similar to annotated OPXV proteins; however, seven predicted AKPV proteins were most similar to proposed proteins in the newly described Murmansk or NY_014 poxvirus genomes: AKPV009, 010, 024, 025, 203, 204, and 205 (Appendix A). Murmansk and NY_014 isolates represent poxviruses that are sister to the OPXV genus but are currently not assigned to a genus [24], similar to Yoka poxvirus [56]. AKPV009/010 are most likely the products of fragmentation of the C-type lectin protein CDS. AKPV009 is similar to the C-terminus of the Murmansk C-type lectin-like protein, while AKPV010 is similar to the N-terminus (Figure 2). Several OPXVs, including CPXV, SKPV, and AKMV encode full length C-type lectin genes, though many OPXV genomes encode shorter C-type lectin genes. For instance, ECTV C-type lectin is also fragmented into two CDS (EVM006 and 007, Figure 2). Interestingly, even though the sequence of AKPV C-type lectin protein is more similar to Murmansk, the start and end points of the predicted genes are more similar to OPXV counterparts (Figure 2).

AKPV024 was most similar to CKM51_gp196 (Ankyrin) in the NY_014 genome, with 92.5% nucleotide identity. AKPV025 was most similar to NY_014 CKM51_gp195 (hypothetical protein), but AKPV025 is much shorter, at 132 nt compared to 552 nt for NY_014 CKM51_gp195. The 38 nt at the AKPV025 N-terminus exhibited 92.1% nucleotide identity with position 30–67 in NY_014 CKM51_gp195, and the AKPV025 C-terminus was 91.5% identical to 393–486, producing an alignment with a 325 bp gap in AKPV025. 

AKPV203, 204, and 205 were most similar to annotated genes in the Murmansk genome. AKPV203 was most similar to poxvirus protein B22R (surface glycoprotein). AKPV204 and 205 were most similar to Murmansk-186 and ankyrin-like protein Murmansk-194 (90.8% and 71.9% amino acid identity), respectively. Murmansk-194 has no known or predicted function, and there is no known homolog to Murmansk-194 or 186 in any poxvirus or other genome. The next best BLAST hit for AKPV205 had <50% amino acid identity and was a truncated ankyrin-like protein in the Yoka poxvirus genome. The next best BLAST hit for AKPV204 also had <50% amino acid identity and was IL-1 beta receptor from AKMV-2013. Interestingly, while AKPV203, 204, and 205 are consecutive in the AKPV genome, the homologs Murmansk-186, 194, and 007 (B22R) are not syntenic and are dispersed in the Murmansk genome. 

Only three of the predicted proteins in the AKPV genome exhibited less than 60% amino acid identity with known poxvirus proteins: AKPV182, 187, and 190. All three genes shared similarity and synteny with poxvirus genes encoding proteins with predicted functions in immune modulation, virulence, and/or host range, including hemagglutinin (AKPV182), EEV type-I membrane glycoprotein VACV-COP-B5R (AKPV187), and poxvirus T4 protein (AKPV190). 

Alignment of AKPV182 (hemagglutinin) with OPXV hemagglutinin homologs revealed an insertion in AKPV192 (Figure 3). Further investigation revealed this insertion contained 11 repeats of “GAAACTCCTGAATCAGTTACTACTCTAGCA”. The reading frame was not disrupted by the insertion, and a predicted transmembrane domain was unaffected. The predicted translated peptide contained 11 repeats of the amino acid sequence “VTTLAETPES”, generating a 120 amino acid insertion. BLAST search of just the 120 amino acid insertion using blastp revealed low level similarity between the repeat region and a portion of adhesive plaque matrix protein-like gene from Mizuhopecten yessoensis (Yesso scallop), with 55% amino acid identity across 117 amino acids of the repeat. BLAST analysis of the nucleotide sequence of the repeat region revealed the highest similarity (72% nucleotide identity, e-value 9e-31) with a region of the cyanobacterium Geminocystis sp. strain NIES-3708 genome corresponding to GM3708_3008 flagelliform silk protein (BAQ62602.1). No putative protein domains were identified in the repeat region based on queries made using the Conserved Domain Database (NCBI), SMART-Simple Modular Architecture Research Tool (EMBL, Universitat Heidelberg), Pfam (EMBL-EBI), Motif Scan, InterPro, ScanProsite – (ExPASy) using a cut-off e-value of 0.01.

Alignments of AKPV187 (B5R) and AKPV190 (T4) with OPXV homologs did not reveal any large insertions or deletions. These genes were analyzed for evidence of selection using PAML. Analysis using the two-ratio branch model revealed that AKPV187 (B5R) was significantly more likely to contain a different dN/dS than reference OPXV B5R homologs. Further investigation using branch-site models (model A and A1) revealed that AKPV187 was more likely to contain sites exhibiting positive selection than the reference B5R homologs (Appendix A). Bayes empirical bayes analysis [44] revealed ten sites that were under positive selection in AKPV187 (probability dN/dS > 1 was greater than 95%) (Appendix A). AKPV190 (T4) was not more likely to contain a different dN/dS than other OPXV homologs, and examination by site-branch model did not reveal any sites indicative of adaptive evolution in AKPV. Across all T4 homologs examined, four sites were identified where probability dN/dS > 1 was greater than 95%. We examined these sites in AKPV and compared which reference sequences shared similar amino acids at those sites. AKPV190 residue 39(L) was shared with RCNV and similar to I in OPVA, MPXV, CPXVA, and CPXVB; residue 174(I) was shared with VPXV and similar to AKMV (L); residue 176(Y) was shared with SKPV; and residue 187(R) was similar to K in CPXV, CMLV, and TATV.

### 3.4. Comparison of Putative Host Range/Virulence Factors

The AKPV genome contained host range/virulence genes typical of the OPXV genus, including homologs of VACV-COP-E3L, K3L, K1L, P28/N1R, B5R, C7L, T4, C3L, CrmB, and serpins SPI-1, 2 and 3 (Appendix A). The AKPV genome, however, did not contain several OPXV genes that have been identified as host factors or virulence genes in other OPXVs. The AKPV isolate genome does not encode a homolog of virulence factor VACV-COP-B7R (bifunctional membrane protein 21 kDA precursor/processed to 18 kDA), a known virulence gene in VACV, where its deletion reduced lesions in a mouse skin model [57]. A frame shift mutation in AKPV results in a premature stop codon after 21 amino acids. Among OPXVs, VARV does not encode a B7R homolog (VARV_SLE_1969, VARV_GNQ_1969, VARV_BRA_1966, VARV_BRA2_1966, VARV_IND_1964, VARV_DRC_1970), and the homolog in CMLV-KAZ-1966 is truncated. The AKPV isolate genome also did not contain a homolog of interleukin-1 receptor antagonist VACV-COP-C4L. A large deletion would result in a protein of only 12 amino acids in length before a stop codon. Among OPXVs examined, the ECTV-Moscow genome also contained a deletion and did not encode a VACV-COP-C4L homolog, and C4L was fragmented in Horsepox virus (HSPV). C4L is not encoded in VARV, MPXV, Rabbitpox virus or VACV Western Reserve strain [58]. Lastly, the AKPV genome did not contain homologs of Type 2 tumor necrosis factor receptor-like (TNFR-II) genes CrmC, CrmD, and CrmE, which are found in CPXV genomes but not most other OPXVs. Similar to other OPXVs, the AKPV genome did not encode an M13L homolog, which is found in “clade II” poxviruses, which are sister to OPXVs, including Tanapox virus, Myxoma virus, Deerpox virus, Swinepox virus and Lumpy skin disease virus [58].

### 3.5. Possible Recombination with Ectromelia Virus (ECTV)

The AKPV genome contained an intact predicted A-type inclusion protein gene (AKPV150) of 3324 nucleotides (1107 amino acids). The A-type inclusion protein was shorter than that from CPXV-B (1279 amino acids), AKMV-2013 (1213 amino acids), and RCNV-Herman (1221 amino acids) due to deletions in the middle of the predicted protein (Appendix A). AKPV150 was most similar to the A-type inclusion protein gene from ECTV, with 94.14% amino acid similarity (BLOSUM62, threshold 1). AKPV150 and ECTV ATI also shared many of the same deletions relative to other OPXV ATI proteins. The ATI gene and neighboring sequence in the AKPV genome exhibited very high similarity to ECTV genome, which led to an investigation into potential recombination between ECTV and AKPV in this region of the genome.

Comparison of proposed AKPV coding regions with annotated genes from other OPXVs revealed AKPV168, 169, and 170 had high similarity to ECTV 140, 141, and 142, with AKPV 169 and ECTV 141 sharing >99% amino acid identity. Due to the high similarity across several sequential genes, a region from AKPV143 (DNA helicase) to AKPV175 (hypothetical protein) of the AKPV genome (corresponding to position 138,271 to 166,556) was examined for recombination. Potential recombination events were investigated using an alignment of this region of the AKPV genome with corresponding regions from AKMV-2010, VARV-BRA, HSPV-MNG, CPXV-A, CPXV-C, CPXV-E3, TATV, CMLV, ECTV-Moscow, ECTV-Hamptead, ECTV-Naval, OPVA, VPXV, SKPV, and RCNV-85A reference genomes. 

Recombination analysis using distance plot and bootscan in RDP4 software identified two potential recombinant regions between AKPV and ECTV (Figure 4, Appendix A). The first region spanned almost 4 kb and corresponded to position 146,066 to 149,930 in the AKPV genome (142,812 to 146,694 in ECTV Moscow), containing the full coding sequence of the A-type inclusion protein (Appendix A) and the C terminus of the DNA-dependent RNA polymerase subunit RPO132 coding sequence. The second potential recombinant region spanned almost 2 kb, from position 161,820 to 163,874 in the AKPV genome (158,512 to 160,458 in ECTV Moscow). This potential recombinant region contained the N terminus of AKPV168 (secreted glycoprotein), AKPV169 (profilin-like protein), AKPV170 (type-I membrane glycoprotein), AKPV171 (hypothetical protein), and the C-terminus of AKPV 172 (hydroxysteroid dehydrogenase).

Based on the results of the recombination analysis using RDP4, AKPV143 to AKPV175 was split into four regions: two predicted recombinant regions, an intermediate region, and a flanking region (Figure 4). Each region was aligned to corresponding sequences from OPXV reference genomes. Nucleotide identity between ECTV and Old World OPXV sequences was very high in control regions 1 and 3, but much lower in putative recombinant regions 2 and 4 (Table 3). Within the putative recombinant regions, AKPV and ECTV sequences were >90% identical.

Phylogenetic analysis of the potential recombinant regions revealed significant rearrangement compared to phylogenies produced from other areas of the genome. When comparing either the core region of the genome (Figure 1) or a region flanking the putative recombinant sites (Appendix A), ECTV clusters with other Old World OPXVs and AKPV forms its own sister branch separated from Old World OPXVs. Phylogenetic trees generated using only the putative recombinant regions place ECTV next to AKPV, and it is no longer part of the tight Old World OPXV clade (Appendix A). This phenomenon is not specific to the ECTV-Moscow strain, and was observed in ECTV-Naval and Haptead strains, which are 100% identical to ECTV Moscow in the suspect recombinant regions. Together, the sequence similarity and phylogenetic trees suggest that ECTV contains an AKPV-like sequence rather than the other way around (Table 3, Appendix A).

## 4. Discussion

The isolation and original characterization of the poxvirus isolated from a patient in Alaska suggested that this isolate represents a novel, divergent OPXV capable of infecting humans. The complete genome sequence of the AKPV isolate provided unique observations when compared with other OPXVs. The large genome and broad host range of *Cowpox virus* has led to the hypothesis that extant OPXVs evolved from a CPXV-like ancestor through gene loss and modification [55,59]. Compared to CPXV-BR, the AKPV genome exhibited loss of 23 genes and gain of six genes, including two genes found in Murmansk poxvirus but not in currently known species of the OPXV lineage.

Phylogenetic analyses and sequence similarity of conserved core genes indicated that AKPV is more closely related to Old World OPXVs than to North American OPXVs, in agreement with previous findings using nine conserved genes [30]. Additionally, most of the predicted proteins in the AKPV genome were more similar to Old World OPXV proteins; however, AKPV was isolated in Alaska. The uncertain geographical range of AKPV along with its significant divergence precludes its inclusion in either the Old World or New World OPXV groups. A few predicted AKPV proteins were more similar to New World OPXVs, including several known virulence proteins. Interestingly, seven predicted proteins were most similar to proteins found in novel poxvirus isolates NY_014 and Murmansk, including a homolog of Murmansk-194, a gene that was previously thought to be unique to the Murmansk genome [24]. Of the seven genes that shared similarity with Murmansk and NY_014, three were consecutive in the AKPV genome; however, these three genes were most similar to genes that were dispersed throughout the Murmansk genome. This suggests that the introduction of genes to/from Murmansk or NY_014 was not likely from a simple recombination event. It is difficult to construct the evolutionary history that could have generated the current state of these genomes, but future isolates may provide insight. The presence of genes in AKPV that were more similar to New World OPXVs, NY_014, or Murmansk poxviruses could represent insight into ancestral forms of shared poxvirus genes. Alternatively, these genes may have been acquired independently or through recombination events followed by selection, especially if they have had an extensive amount of time to diverge. The origin, geographic location and natural reservoir of NY_014 and Murmansk remain a mystery, much like that of AKPV. Future investigation, possibly including as yet undiscovered isolates or new poxvirus species may shed light on the relationship between these viruses.

The success of a poxvirus infection depends on the virus’s ability to evade the host immune response. Several genes in the poxvirus genome are known to play a role in modulating the host antiviral response. It has been hypothesized that the presence or absence of such genes as well as sequence differences may underlie the host range and virulence of a given poxvirus. In general, these genes are located at the terminal regions of the genome and often exhibit lower sequence identity and lineage-specific distribution [1,58]. The host range/virulence gene repertoire of AKPV was typical of OPXVs. Three host range/virulence genes exhibited notably low sequence identity with other OPXV counterparts: T4, B5R and hemagglutinin. Examination of B5R revealed evidence of possible selection in the AKPV lineage based on dN/dS analysis. Investigation of the hemagglutinin gene revealed a novel 120 amino acid insertion. The insertion did not create a frame shift or affect the predicted transmembrane domain; however, it is possible that it could lead to functional differences. A study in VACV and ECTV identified hemagglutinin may pay a role in the activation of Natural Killer Cells during poxvirus infection [60]. Investigation of the novel hemagglutinin gene from AKPV may reveal whether the insertion affects protein folding, stability or function during infection. Lastly, the AKPV genome did not contain several OPXV host range/virulence genes, including C4L, B7R, CrmC, CrmD, or CrmE. These changes in gene sequence and gene repertoire may influence the host range and virulence of AKPV, but host range and virulence cannot be predicted by gene content or sequence similarity. Future studies are required to better understand this novel virus, for which very little is known. 

Careful analysis of the AKPV isolate genome revealed unexpected recombination with ECTV. ECTV was first identified in 1930 in a laboratory mouse and is the causative agent of mousepox, a disease found in mouse colonies in Europe, Asia, and the Americas [61,62]. Although the distribution and natural reservoir of ECTV remains unknown, it is suspected to circulate among wild rodents in Europe [62]. The geographic distribution of AKPV is also unclear and the source of exposure to the Alaska resident has not yet been resolved [30]. While concrete data about the distribution of either species is largely lacking, AKPV was isolated in the Americas and ECTV is thought to circulate in Europe. Given the current estimates of when Old and New World OPXVs separated, the high sequence similarity in suspect recombinant regions between AKPV and ECTV suggests either a short time since recombination or very low rate of evolution for this region of the genome. 

Taken together, this study highlights the need for future studies of natural poxvirus circulation in wildlife to generate a better understanding of OPXV ecology and better preparedness for zoonotic infections in humans. The discovery of new, divergent OPXVs introduces the opportunity to re-evaluate the *Orthopoxvirus* genus. The discovery of new diverse species can strengthen existing understanding, and provide further insight or improve the resolution of previous analyses. It is very likely that the wealth of sequencing information reflecting OPXV diversity will continue to increase. This information can be used to inform diagnostics and may provide increasingly more accurate information about OPXV evolution and origins.

## Figures and Tables

**Figure 1 viruses-11-00708-f001:**
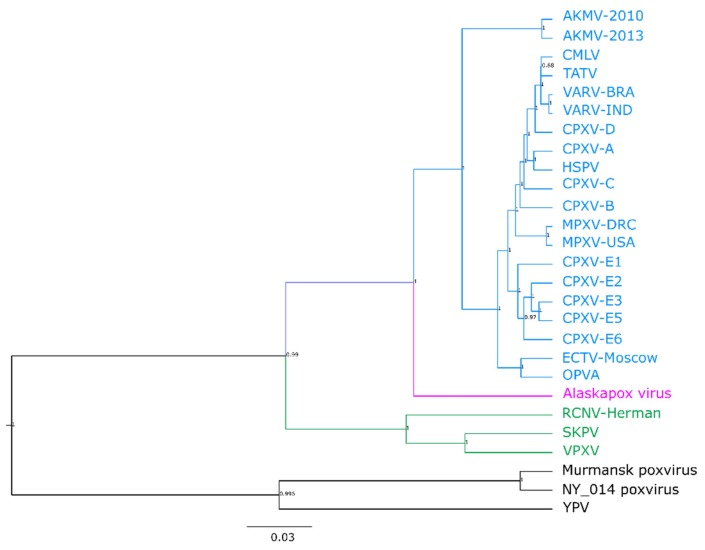
Phylogenetic analysis of the conserved central region of the Alaskapox virus (AKPV) genome with representative OPXVs, Murmansk poxvirus, NY_014 poxvirus, and Yoka poxvirus (YPV). Twenty-seven poxvirus sequences (Appendix A) were aligned using MAFFT. Old World OPXVs are highlighted in blue; New World OPXVs are highlighted in green; AKPV is highlighted in pink. Phylogenetic tree was estimated using BEAST v. 2.5.1 using the GTR+G+I model under a relaxed lognormal molecular clock. Posterior probability is shown next to each node.

**Figure 2 viruses-11-00708-f002:**
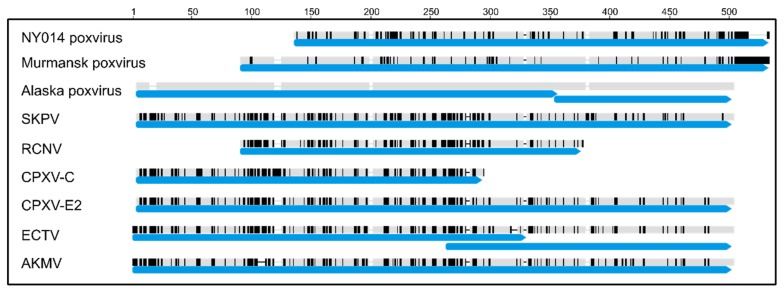
Alignment of C-type lectin genes from AKPV and reference OPXVs. Nucleotide identity to AKPV is shown by black or gray shading: identities are shown in gray, differences in black. Black or gray bars indicate sequence, whereas black or gray horizontal lines indicate gaps. Blue arrows indicate the locations of annotated CDS. Alignment position is shown above the graph.

**Figure 3 viruses-11-00708-f003:**
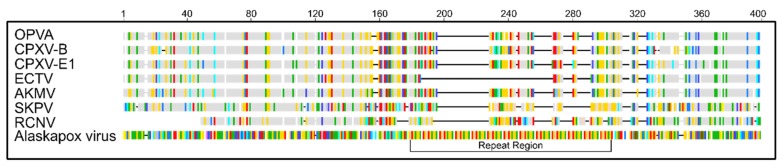
Comparison of predicted hemagglutinin protein from Alaskapox virus (AKPV) and reference OPXVs. Amino acid sequence of AKPV hemagglutinin is shown in RasMol color scheme. For others, amino acid sequence is colored compared to AKPV protein: identities are shown in gray; differences are colored according to RasMol scheme. Gaps not shared with AKPV are shown by black horizontal lines. Sequences from OPVA, CPXV-B, CPXV-E1, ECTV-Moscow, AKMV-2013, SKPV, and RCNV-85A are shown.

**Figure 4 viruses-11-00708-f004:**
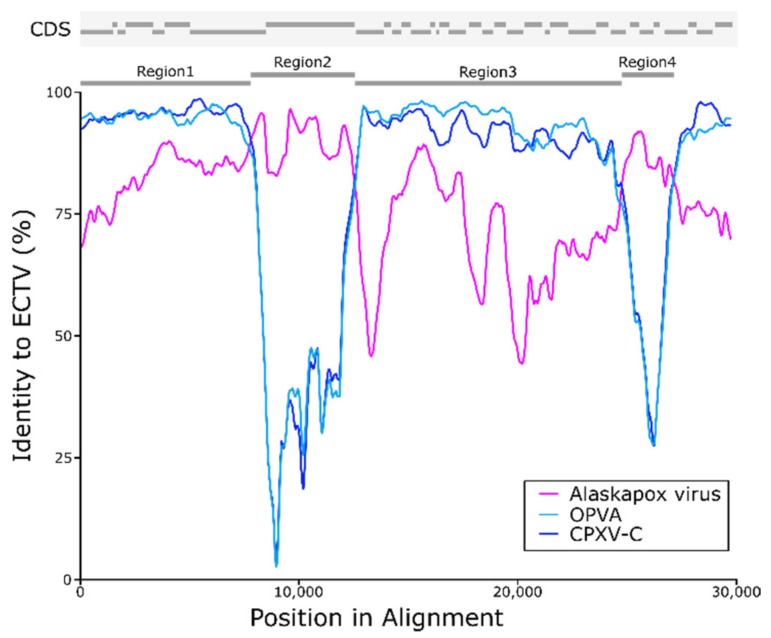
Potential recombination between Ectromelia virus (ECTV) and the Alaskapox virus (AKPV) isolate. Similarity plot displaying the sequence identity of AKPV, Orthopoxvirus Abatino (OPVA) and Cowpox virus (CPXV-C) to ECTV (Moscow strain) in the putative recombination region. Across most positions examined, OPVA and CPXV-C have higher similarity to ECTV than AKPV. However, the AKPV sequence has higher sequence identity than ECTV in two regions that can be seen as the pink AKPV line crossing the blue OPVA and CPXV lines. Above the graph, there is a schematic showing locations of annotated coding sequences (CDS), putative recombinant regions 2 and 4, and control regions 1 and 3.

**Table 1 viruses-11-00708-t001:** Average percent nucleotide identity of the conserved core region of the Alaskapox virus isolate (AKPV), representative Cowpox virus (CPXV), Akhmeta virus (AKMV), Old World OPXV (Old World), New World OPXV (New World) and Yoka poxvirus (YPV) genomes (Appendix A). Gaps were not included in this analysis.

	CPXV	AKMV	Old World	New World	YPV
AKPV	92.93	93.52	92.87	87.10	74.88
CPXV	98.44	95.62	97.52	87.38	74.78
AKMV		99.39	95.48	87.27	74.72
Old World			97.52	87.29	74.79
New World				92.02	75.58

**Table 2 viruses-11-00708-t002:** Percent amino acid identity of TNFα receptor-like AKPV011 and ankyrin repeat-containing AKPV013 with corresponding proteins from representative OPXV genomes, including SKPV, RCNV-Herman, CPXV-E6, ECTV-Moscow, and AKMV-2013.

	Amino Acid Identity	
	AKPV011	AKPV013
SKPV	93.00	90.74
RCNV	92.50	92.42
CPXV	90.00	88.50
ECTV	87.13	89.40
AKMV	86.57	80.05

**Table 3 viruses-11-00708-t003:** Size and nucleotide identity of putative recombination sites in Ectromelia virus (ECTV) genome. Percent nucleotide identity to ECTV-Moscow is shown for Alaskapox virus isolate (AKPV), AKMV-2010, OPVA, and RCNV-85A. Regions 2 and 4 refer to suspect recombination sites; regions 1 and 3 are neighboring genomic regions, as shown in Figure 4. Gaps were not removed for this analysis.

	Length	AKPV	AKMV	OPVA	RCNV
Region 1	7794	92.99	95.41	97.95	87.21
Region 2	3882	94.50	76.11	67.55	64.35
Region 3	11,816	86.50	91.34	95.59	75.15
Region 4	1946	91.93	82.59	75.66	63.35

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
