# Peer review of "Genome of Alaskapox Virus, a Novel Orthopoxvirus Isolated from Alaska"

_viruses, 2019, doi:10.3390/v11080708_

Round 1

Reviewer 1 Report

This paper describes the genomic characteristics of a recently discovered zoonotic poxvirus that was isolated from a human patient in Alaska and proposes its classification as a new species (Alaskapox virus) highly related to, but distinct from, both the New and Old world orthopoxviruses. The analyses appear to be appropriate and the conclusions drawn are sound.

One of the more interesting findings is the presence of two genes which appear to be similar to two found so far only in the Murmansk poxvirus which lies outside the current orthopoxvirus lineages. The authors, quite rightly, state that much is still to be learned about the natural hosts, origins and evolution of poxviruses (particularly in wildlife), and seem to be suggesting a common poxviral ancestor may explain the presence of these genes (and others) in the Alaskapox and Murmansk genomes. However, they may wish to comment on whether or not independent acquisition of these genes and subsequent adaptation (potentially host driven) may be an alternative explanation.

The authors also say that the source of the original human infection with Alaskapox is unknown. Can the authors draw any conclusions about the possible natural host based on the similarity of some of the virulence genes such as the TNF-receptor-like gene or IFN-gamma receptor-like protein with the known sequences from mammalian hosts?

In lines 445-448 where the authors discuss “…changes in gene sequence and gene loss…” can they be certain that the Alaskapox genome has lost these genes in each case?

The paper is well written and I have very few comments to make, but there are one or two areas where a bit of clarification is required;

line 93 “ab inito” – should this be “ab initio”? Also does this refer to a software package or particular process? If this is the case then this should be referenced   

line 175-176 I am unclear if the numbering (61-80) refers to the concatamer resolution sequence within the Alaskapox genome or the VACV-Cop genome?

line 419 insert the word “not” after the word “but”

Author Response

Comments and Suggestions for Authors

This paper describes the genomic characteristics of a recently discovered zoonotic poxvirus that was isolated from a human patient in Alaska and proposes its classification as a new species (Alaskapox virus) highly related to, but distinct from, both the New and Old world orthopoxviruses. The analyses appear to be appropriate and the conclusions drawn are sound.

One of the more interesting findings is the presence of two genes which appear to be similar to two found so far only in the Murmansk poxvirus which lies outside the current orthopoxvirus lineages. The authors, quite rightly, state that much is still to be learned about the natural hosts, origins and evolution of poxviruses (particularly in wildlife), and seem to be suggesting a common poxviral ancestor may explain the presence of these genes (and others) in the Alaskapox and Murmansk genomes. However, they may wish to comment on whether or not independent acquisition of these genes and subsequent adaptation (potentially host driven) may be an alternative explanation.

We thank the reviewer for this suggestion. We have added language to the discussion to highlight this idea. 

The authors also say that the source of the original human infection with Alaskapox is unknown. Can the authors draw any conclusions about the possible natural host based on the similarity of some of the virulence genes such as the TNF-receptor-like gene or IFN-gamma receptor-like protein with the known sequences from mammalian hosts?

We feel this request is outside the scope of the current paper. The complete genome sequence will be made publically available for interested parties to explore questions such as this.

In lines 445-448 where the authors discuss “…changes in gene sequence and gene loss…” can they be certain that the Alaskapox genome has lost these genes in each case?

“Gene loss” changed to the more appropriate “gene repertoire” to address this comment.

The paper is well written and I have very few comments to make, but there are one or two areas where a bit of clarification is required;

line 93 “ab inito” – should this be “ab initio”? Also does this refer to a software package or particular process? If this is the case then this should be referenced   

Changed to “ab initio” and reference was added, per the reviewer’s request.

line 175-176 I am unclear if the numbering (61-80) refers to the concatamer resolution sequence within the Alaskapox genome or the VACV-Cop genome?

This has been clarified.

line 419 insert the word “not” after the word “but”

This has been done. We thank the reviewer for his/her time.

Reviewer 2 Report

It is the first full genome sequencing of the Alaskapox virus

The authors did a great effort to assemble the whole genome using the next generation sequencing and compare it to the whole genome of other poxviruses. 

Despite the similarities to the other orthopoxviruses, the Alaskapox virus lacked 23 genes, two them are functional genes, which is a very important observation. 

An interesting finding that require more in-depth research is the homology of the Alaskapox virus to old wold orthopoxviruses in spite of existing partial homology to two new world poxviruses. 

The manuscript showed the importance of performing more research in the poxviruses field as after the eradication of small poxvirus, little has been done.

Author Response

Comments and Suggestions for Authors

It is the first full genome sequencing of the Alaskapox virus

The authors did a great effort to assemble the whole genome using the next generation sequencing and compare it to the whole genome of other poxviruses. 

Despite the similarities to the other orthopoxviruses, the Alaskapox virus lacked 23 genes, two them are functional genes, which is a very important observation. 

An interesting finding that require more in-depth research is the homology of the Alaskapox virus to old wold orthopoxviruses in spite of existing partial homology to two new world poxviruses. 

The manuscript showed the importance of performing more research in the poxviruses field as after the eradication of small poxvirus, little has been done.

We thank the reviewer for his/her time.

Reviewer 3 Report

Very nice genetic exploration of an apparently novel poxvirus.  The only suggestion I have is to ask you to please revise wording of Lines 409-410 in the discussion.  It is unclear which genes you are referring to at the end of the sentence when you compare to OPXV genes.  The sentence currently reads "Compared to CPXV-BR, the AKPV genome exhibited loss of 23 genes and gain of six genes, including two genes found in Murmansk poxvirus but in currently known species of the OPXV lineage."

Author Response

Comments and Suggestions for Authors

Very nice genetic exploration of an apparently novel poxvirus.  The only suggestion I have is to ask you to please revise wording of Lines 409-410 in the discussion.  It is unclear which genes you are referring to at the end of the sentence when you compare to OPXV genes.  The sentence currently reads "Compared to CPXV-BR, the AKPV genome exhibited loss of 23 genes and gain of six genes, including two genes found in Murmansk poxvirus but in currently known species of the OPXV lineage."

This wording has been revised. We thank the reviewer for his/her time.